

# Automatic analysis of treadmill running to estimate times to fatigue and exhaustion in rodents

Dmitry V. Zaretsky, Hannah Kline, Maria V. Zaretskaia and Daniel E. Rusyniak

Department of Emergency Medicine, Indiana University School of Medicine, Indianapolis, IN, USA
Department of Pharmacology and Toxicology, Indiana University School of Medicine, Indianapolis, IN, USA

## ABSTRACT

**Introduction.** The determination of fatigue and exhaustion in experimental animals is complicated by the subjective nature of the measurement. Typically, it requires an observer to watch exercising animals, e.g. rats running on the treadmill, and to identify the time of the event. In this study, we hypothesized that automatic analysis of the time-averaged position of a rat on a treadmill could be an objective way for estimating times to fatigue and exhaustion. To test this hypothesis, we compared these times measured by a human observer to the results of an automated video tracking system.

**Methods.** Rats, previously familiarized to running on the treadmill, ran at a fixed speed with zero incline, until exhaustion. The experiments were performed at either room temperature (24 °C) or in a hot environment (32 °C). Each experiment was video recorded. A trained observer estimated the times to fatigue and exhaustion. Then, video tracking software was used to determine the position of the animals on the treadmill belt. The times to fatigue and exhaustion were determined, based on the position on the treadmill using predefined criteria.

**Results.** Manual scores and the average position on the treadmill had significant correlation. Both the observer and the automated video tracking determined that exercise in a hot environment, compared with the exercise at room temperature, results in shorter times to exhaustion and fatigue. Also, estimates of times made by the observer and the automated video tracking were not statistically different from each other.

**Discussion.** A similarity between the estimates of times to fatigue and exhaustion made by the observer and the automated technique suggests that video tracking of rodents running on a treadmill can be used to determine both parameters in experimental studies. Video tracking technique allows for a more objective measure and would allow for an increased performance in experimentation. The Supplemental information to this manuscript contains an Excel file, which includes the code in Virtual Basic with freeware license, to process and visualize running data and automatically estimate the times to fatigue and exhaustion. Instructions for the software are also included.

Corresponding author
Dmitry V. Zaretsky,
zaretsky.dv@hotmail.com,
diza_in@hotmail.com

# INTRODUCTION

Fatigue and exhaustion are a part of everyday life. For athletes, the ability to overcome them can be the difference between winning and losing an endurance event. A similar
need for thwarting fatigue and exhaustion extends to military personnel, who must remain alert to avoid jeopardizing their missions (*Hursh et al., 2004*; *Shattuck & Matsangas, 2015*). Problems with fatigue and exhaustion, however, extend beyond the healthy individual. Cancer patients and persons with a variety of chronic illnesses, e.g., arthritis, multiple sclerosis, consistently state that fatigue dramatically decreases their quality of life (*Bang et al., 2015*; *Gossec et al., 2014*; *Patejdl et al., 2016*; *Winger et al., 2015*). However, the lack of understanding of the central and peripheral mechanisms underlying fatigue and exhaustion hampers the development of new treatments. This growing issue in both healthy and chronically ill individuals creates a clear need for research into its mechanisms. While fatigue and exhaustion are recognized as significant health issues, they are difficult to diagnose and treat, in part due to their subjectivity (*Heckman, Mathew & Carpenter, 2015*; *Jason et al., 2015*). Both terms, ''fatigue'' and ''exhaustion'', often are used interchangeably (*Hasegawa et al., 2008*; *Walters et al., 2000*). In fact, the Merriam-Webster dictionary defines fatigue as ''weariness or exhaustion from labor, exertion, or stress'' (*Merriam-Webster.com, 2018*). However, many researchers separate them by stating that exhaustion is synonymous with fatigue but is more intense. Cancer patients who are fatigued experience difficulty concentrating, anxiety, and a gradual decrease in stamina. People who suffer from exhaustion report frank confusion that resembles delirium, emotional numbness, sudden loss of energy, and difficulty in staying awake (*Saligan et al., 2015*). Fatigue indicates declining ability to respond to stressors, while exhaustion indicates almost complete inability to respond to stressors (*Olson et al., 2008*).

Based on the previous definitions of fatigue and exhaustion, exercise (e.g., treadmill, swimming) is a common model used to study it. In the animal model of exertional fatigue, the repeated slipping from front running position to the middle is a reflection of the inability to maintain optimal physical performance, and will be consistent with fatigue. The rat always restores front running position after receiving electric shocks, which in turn, corresponds to the ability to respond to stressors. However, with time, the ability of stress to restore running performance declines, and repeated shocks do not help rat to keep pace with the moving belt. This stage phenomenologically corresponds to an inability to respond to stressors. The status, in which the animal no longer responds to prodding to keep with the moving belt, is translationally similar to the exhaustion. Here, the major feature of exhaustion is an inability of animal to continue to exercise despite prodding. In treadmill experiments, when an animal is nearing exhaustion, it has a sustained inability to keep the pace with the belt. This means that the animal is permanently located near the back of the treadmill, and unable to make it back to the front running position despite any unpleasant or painful stimuli such as electric shocks. The definition of exhaustion, therefore, could be defined exclusively by the position on the treadmill. In contrast, the definition of fatigue is usually more complex than just the position of the animal on the treadmill. To make it translationally valid, the researchers usually include clearly recognizable and characteristic behavioral pattern that reflects the loss of the optimal physical performance. If this pattern is observed for more than some predefined period of time, it is considered as an objective sign of fatigue (*Fuller, Carter & Mitchell, 1998*). This behavior is also accompanied by the animal having transient periods when it cannot keep pace with the treadmill so that the

PeerJ ________________________________________________

animal moves toward the back of the treadmill, may receive a prodding stimulus, and then returns to the front of the treadmill. This transient inability to keep pace with the treadmill is used by some authors as the definition of fatigue (*Rodrigues et al., 2009*; *Soares et al., 2004*). The recurrent pattern of an animal sliding to the back of the treadmill, followed by a return to the front of the treadmill would result in the average position of the rat on the belt of the treadmill shifting from the front to the middle.

Based on the above, we proposed that in animal experiments the average position of the animal on the treadmill can be measured and used to define fatigue and exhaustion. Fatigue would correlate with the shift of average position at the treadmill from the front to the middle of the lane and exhaustion to the back of the lane. To test this approach, we recorded the average position of rats running on a treadmill at room temperature and in hot environment. We compared results of obtained by a human observer with that obtained using video tracking software.

## METHODS

### Animals

Male adult Sprague-Dawley rats (weight $300 \pm 20$ g; Harlan, Indianapolis, IN, USA) were used in this study. All procedures were approved by the Indiana University Animal Care and Use Committee (#10093). Experiments were performed using single-housed rats that were maintained in a 12 h light/dark cycle and fed ad libitum. We conducted experiments on fully conscious rats between the hours of 10:00 a.m. and 4:00 p.m. The experiments were performed in a custom-made environmental chamber, which housed a treadmill.

### Experimental protocol

To validate the use of automatic video tracking software for estimation of the times to fatigue and exhaustion, we had two groups of rats ($N = 6$ each) run until exhaustion at either room temperature (ambient temperature $24 \pm 1\,°C$, humidity 30–70%) or in a hot environment (ambient temperature $32 \pm 1\,°C$, humidity below 20%). Initially, the home cage with the rat was placed outside the environmental chamber at room temperature. The rat was allowed to adapt in the home cage before being transferred to the treadmill lane.

The rats were subjected to constant-speed exercise, and ran until exhaustion at a fixed treadmill speed of 18 m/min with zero incline. The runs were video recorded and simultaneously manually scored by one of the authors (HK). Video files were analyzed using video tracking software Anymaze (Stoelting Co., Wood Dale, IL, USA) by a researcher blinded to the results of the manual scoring. Manual and automated scores were compared. Automated scores were used to determine times to fatigue and exhaustion in further experiments.

### Treadmill running
#### *Familiarization to treadmill running*
Prior to all treadmill experiments, the rats were familiarized to running on a motorized rodent treadmill (Columbus Instruments, Columbus, OH, USA) set with zero incline. On the first day of familiarization, the rats were placed on the treadmill with the belt

speed on standby. They were given 10 min to explore their surroundings. After this, the treadmill speed was set to 6 m/min for 5 min. For next four days, the rats were subjected daily to five-minute sessions that included running at progressively increased speeds up to a maximum of 18 m/min by day 5. Previous studies have shown that the workloads of familiarization protocols accustom rats to treadmill running but do not induce training adaptations (*Lambert & Noakes, 1989*). Mild electric stimuli (1 mA, 3 Hz) at the back of the treadmill chamber promoted the learning of running behavior.

### Treadmill running

On the day of the experiment, the cage with the animal was brought to the research lab. The rats were allowed to acclimate to the new environment for at least 60 min. The temperature of the custom-made environmental chamber, which housed the treadmill, was set to either room temperature or a high temperature.

After the period of acclimation, the rat was placed on the treadmill lane. The treadmill opening was closed with a terminal plate that contained a fan providing air circulation from the chamber, and the belt was turned on at 18 m/min with zero incline. When a rat was consistently unable to maintain the pace of the treadmill and received three continuous electric shocks (1 mA, 3 Hz) on the grid at the back of the treadmill, the belt was stopped. The rat was taken immediately from the lane and was transferred back to the home cage at room temperature.

### Video recording of treadmill experiments

Video recordings were obtained using a USB-camera connected to a laptop running Xtrasense software (surveillance computer-based system, XmediaStudio, Dawang group) using a proprietary codec. The timer of the computer was embedded into the video stream, which was set at the resolution 640 × 480. The analysis can be performed as described using recordings with lower resolutions of 352 × 288 or 320 × 240. Files were stored until analysis could be done.

The video camera was positioned to capture the entire treadmill lane (Columbus Instruments Inc., Columbus, OH, USA) from the side; the camera could also be positioned looking down from the top of the treadmill. To increase contrast between the animal and the background, the opposite face of the treadmill lane was covered from the outside with dark paper. The same effect could be achieved by recording a transparent treadmill lane positioned in front of a black wall.

## Data analysis
### Manual scoring of the quality of treadmill running

An observer scored the quality of the animal's running every minute using a previously described technique (*Guasch et al., 2013*) with minor modifications. The animal was scored a 5 if it ran the majority of a 1-min epoch in the front of the treadmill (proximal third). The rat with this score had a normal running gait with the tail raised. As animals became fatigued they would lay down flat on the belt, and ride to the back of the treadmill. As they approached the back of the treadmill and before they touched the shock grid, they would right themselves and run to the front of the treadmill. This cycle would then repeat.

When the animal spent the majority of the time in a 1-min epoch in this fashion, they were scored a 3. This behavior that we call "belt-riding" resulted in an average position in the middle third of the treadmill. Finally, the animals lost the ability to return to the front of the treadmill after riding to the back. If this was consistent during the time epoch, the average position would be in the low third of the treadmill length, and the animal was scored 1. When the animal refused to run, or was not able to avoid repeated shocks, the score was 0.

Similar to other researchers, we defined the time to fatigue as the time until an animal became unable to continuously keep pace with the treadmill and stay in the front (*Rodrigues et al., 2009*; *Soares et al., 2004*). In our scoring system, this corresponds to the time the animal transitioned from a running score of 5 to 3. To assure that this was not a transient phenomenon, but rather represented true fatigue, a single score of 3 immediately followed by a recovery to score 5 did not qualify as fatigue. Only three consecutive scores of 3 or below counted as fatigue. Exhaustion in our scoring system was defined by a score of 0.

### Analysis with Anymaze

To quantitate the running quality and to automate the analysis, we used video tracking system software (Anymaze, Stoelting Co., Wood Dale, IL). We created an experimental protocol based on the built-in function to measure the distance of the animal's center from a predefined reference point, which for our experiments was defined as the back of the treadmill. The video file was selected as the source of the video stream. The observation area ("apparatus" in Anymaze) was defined in the configuration as required, and the length of the treadmill was calibrated where appropriate. The analysis of the video file was performed, and the data were exported as a file in comma separated values format. The resulting file was imported into Microsoft Excel.

### Data processing using Excel template with the code in virtual basic

Video tracking generates a variable number of data points per unit of time. On a typical laptop, Anymaze is able to estimate 3–8 positions per second, which is equivalent to 10,000–30,000 pairs of (time; position) per hour. At room temperature, a non-trained rat can run up to 90 min at the speed and incline used in our experiments. To process such large datasets, and to obtain averages over selected time intervals (e.g., 1 min), we programmed and used a template in Excel, which is included as Supplemental Information to this manuscript and can be used as freeware.

The difference between actual recording time and time during video play was corrected by a coefficient which was calculated by comparing the elapsed time on video player timer during playback and the timer embedded into the video stream. This coefficient was stable within each recording and ranged from 0.9 to 1.33. More details and the method to calculate the coefficient is provided in the manual for the template.

Initially, the template calculated only averages. The user could select the time interval for the averaging. We then added graphic representations. Finally, we implemented the algorithm to estimate the time to fatigue and the time to exhaustion using the rules similar to ones used by the observer. Time to fatigue was defined as the earliest time when the average animal position over temporal step fell below upper threshold and did not recover

for predefined number of steps. Unless noted, temporal step was 1 min, and predefined number of steps was 3. Similarly, the time to exhaustion was defined as the earliest time when the average animal position over temporal step fell below lower threshold and did not recover for at least a predefined number of steps. Unless noted, the upper threshold was 0.7 and the lower threshold was 0.35.

A detailed user manual for the template is included into the Supplemental Information. Instructions on how to use the template are in the sheet of the Excel file, and are included as a separate file in the Supplemental Information.

### Correlation of scoring by human observer and video tracking software

The one-minute averages of the normalized position on the treadmill calculated by video tracking software (scores ranged from 0 to 1) and manual scoring (scores ranged from 0 to 5) were aligned in columns in an Excel spreadsheet. For each rat we calculated a coefficient of correlation between the manual score and the normalized position obtained by video tracking software. To determine the applicability of this method in different environments, we compared the video tracking software data with manual scoring for rats running at both room temperature and in a hot environment.

To fully automate the quantification of fatigue and exhaustion, we used a least-square test to determine the boundaries between zones which will best correspond to scores 5, 3, and 1 in the manual scoring for each rat. To do that, we set two adjacent cells in MS Excel (D2 and D3 in the Fig. 1) filled with our initial estimates of what the boundaries would be: the upper boundary between running zones 3 and 5 was set at 0.7, while the lower boundary between running zones 1 and 3 was set at 0.35. Then, two extra columns were added for the data for each rat. In the first extra column (E in the Fig. 1) the automated running score was calculated using the estimated boundaries between running zones. The second column (column F, Fig. 1) contained squared difference between the manual and the automated score. The target cell (F2) contained a sum of differences for all running times. By changing the cells containing upper and lower boundaries (D2 and D3) we minimized the value in the target cell (F2). Unfortunately, due to the discrete nature of the target function, algorithms from the Solver tool-kit were ineffective. To our advantage, the function allows for minimizing the target cell by sequentially finding the best estimate for the first variable, followed by determining another variable. We restricted the precision of finding the estimates to two digits after the comma. The estimates of the boundaries between the zones for each rat were averaged.

For processing experimental data, we selected 0.7 and 0.35 for the upper and lower boundaries, accordingly. Using these preset values, we calculated the correlation coefficients for each rat between manual score values and automatic score values. Data was averaged for rats running at both room temperature and in a hot environment.

### Statistical procedures and graphing

Statistical analysis and graphing were performed using Microsoft Excel 2010 (Microsoft, Redmont, WA, USA) and Prism 4.0 (GraphPad Software, San Diego, CA, USA). The comparisons between groups were performed using Student's $t$-test or the analysis of

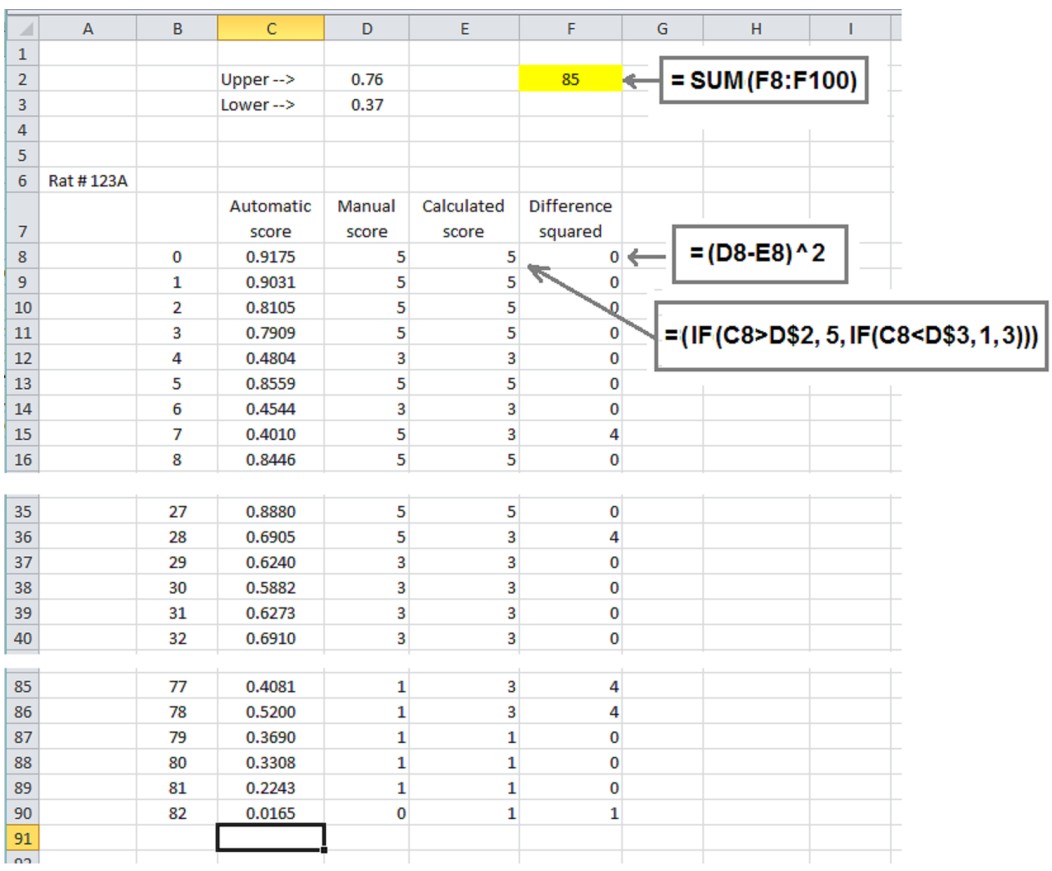

**Figure 1 The screenshot of an example calculating optimal borders between zones for one rat.** Portions of a spreadsheet corresponding to the beginning (rows 8–16), the middle (rows 35–40) and the end of run (rows 85–90) are shown. Rectangles contain the formulas to calculate corresponding cells.

variance (ANOVA), followed by Fisher's least significant difference (LSD) post hoc test, where appropriate. Significance was defined as a $p$-value $<0.05$. Values are presented as means $\pm$SEM.

## RESULTS

### Video tracking software provides an objective measure of running quality

The automatic tracking analysis of the video recording by Anymaze creates a continuous measure of a rat's position on the treadmill relative to the back of the treadmill (Fig. 2A). The length of the treadmill belt is 0.4 m; however, due to the length of rat itself, the range of possible positions was between 0.1 m and 0.33 m. These minimal and maximal values are estimated from the completed recording and were used to calculate the normalized position on the treadmill (with values between 0 and 1).

At the start of exercise, rats run at the front of the treadmill. The automated scoring of individual runs captures more temporal details of the changes in the rat's movement

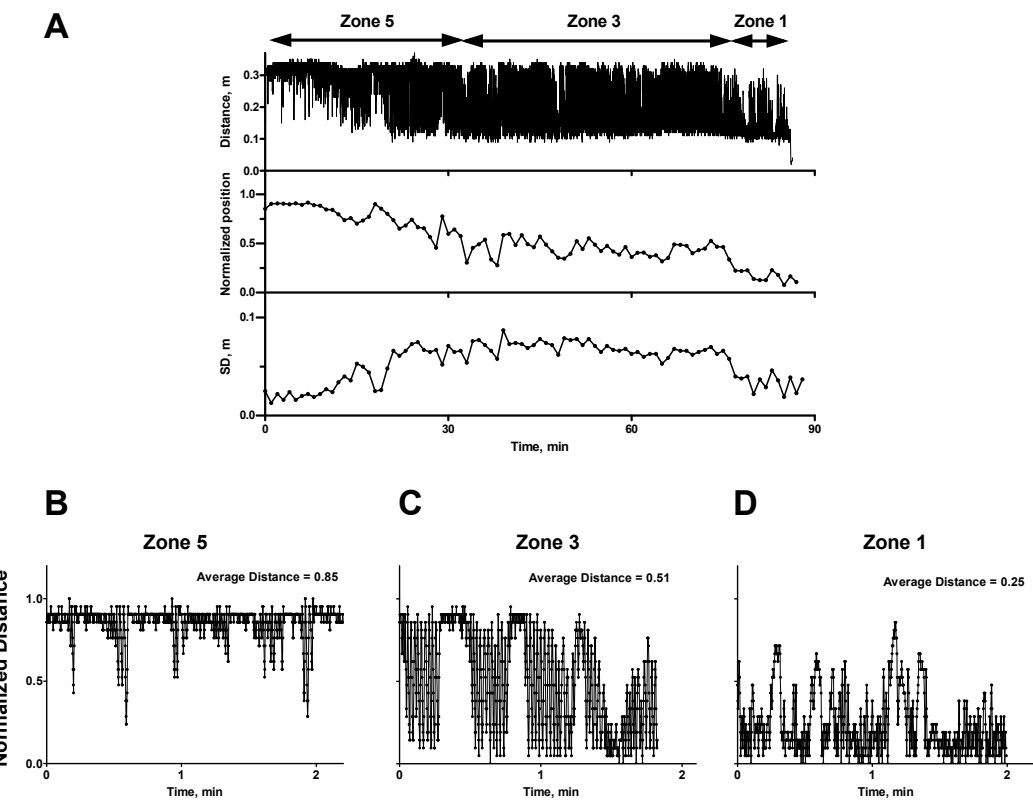

**Figure 2** **Typical result of automatic tracking of a rat running on the treadmill.** (A) The record of complete run. Distance, m-original data; normalized position-data averaged every minute; SD, m-standard deviation for the position for each minute. (B, C, D) Scaled portions of the recording representing rats running with scores 5 (B), 3 (C), and 1 (D) with corresponding average distance for shown time interval.

compared to manual scoring. For instance, recordings often show that before fatigue or exhaustion occurs, some rats transiently appear at the back of the treadmill; in most of these cases they appear to be exploring. In the first 15 min of recording, the lowest points on the curve do not fall below 0.15 m. This means that early on during exercise, when an animal slides back, it does not reach the electric grid and is readily able to recover its position at the front of the treadmill (Fig. 2A, Zone 5). Animals running in Zone 5 have a low variability of position (Fig. 2A, SD graph). Over time, as fatigue develops, there is a change in running quality (that we referred as "belt riding"). Due to more time spent closer to the back, the average position slips to the middle of treadmill or 0.5 in relative value (Zone 3). Zone 3 running is characterized by the highest variability in the animal's position (Fig. 2C). Finally, when the rat approaches exhaustion, it is incapable of reaching the front of the treadmill; it often fails to keep up with the treadmill while positioned near the back of the belt. This corresponds to an automated average position that falls below 0.5. This Zone 1 running also has decreased variability (Fig. 2D).

Reaching the averaged position around zero or below zero corresponds to the exhaustion determined by the observer.

## Manual and automated running scores correlate

Figure 3 compares the results of manual scoring for animals to the results of videotracking. Manual scoring has obvious visual similarity with the results of automatic analysis for experiments performed at room temperature (Fig. 3A) and in hot conditions (Fig. 3B). Manual scores and average positions correlated significantly with an average coefficient of $0.75 \pm 0.03$. The time to reach averaged position close to zero was exactly the same as the time of exhaustion determined by the observer in the case of each rat.

Based on these results, we used the position on the treadmill to calculate the time to fatigue and exhaustion. Using the method of least squares (see 'Methods') we defined the boundaries that correspond to subjective scores of 1, 3 and 5 for each rat (named Zones 1, 3, and 5, correspondingly). Using such boundaries would result in an automatic score closest to the manual score for this rat. On average, the boundary between Zones 5 and 3 appeared at $0.73 \pm 0.02$, and between Zones 3 and 1 at $0.35 \pm 0.04$. This is not surprising as the direct application of instructions to the observer implies the upper boundary at 0.67, and the lower boundary at 0.33.

For further calculations, we selected 0.7 and 0.35 for the boundaries between Zones 3 and 5, and 1 and 3, accordingly. In some individual rats the average values fell below 0.7 for a short period of time, and then scores quickly returned to above 0.7 for prolonged time. In these situations, the posture of the rat was typical for pre-fatigue status. To avoid underestimation of time to fatigue, we introduced an additional rule similar to that used in manual scoring (*Guasch et al., 2013*): time to fatigue was determined when the average position for one minute fell below 0.7 and did not recover for at least three minutes.

## Exercise in a hot environment results in faster fatigue and exhaustion

In rats familiarized to treadmill running, but otherwise untreated, the estimates of time to fatigue made by a human observer and video tracking were not statistically different. At room temperature, fatigue developed after $35 \pm 4$ min according to the human observer or after $30 \pm 3$ min according to the video tracking ($p > 0.05$ when two approaches were compared). In a hot environment, fatigue occurred statistically significantly earlier according to both the human observer and the video tracking ($22 \pm 2$ min for both, $p < 0.05$ when compared to time to fatigue at room temperature). The coefficient of correlation between the times to fatigue determined by the observer and the ones determined by video tracking was 0.79.

Further dynamics differed dramatically between the two environments. At room temperature, exhaustion developed at $88 \pm 5$ min, which was $58 \pm 5$ min after fatigue if determined by the observer or $52 \pm 2$ min if determined by the video tracking ($p > 0.05$). In contrast, in the hot environment, exhaustion occurred at $42 \pm 2$ min (only $21 \pm 2$ min after the development of the fatigue according to both the observer and the video tracking, both values are statistically different from those at room temperature, $p < 0.001$).

The difference between the time to exhaustion and the time when the average position dropped below 0.35 was $4 \pm 1$ min or $7 \pm 1\%$ of the time to exhaustion. The percentage was not different between two environments. The coefficient of correlation between the

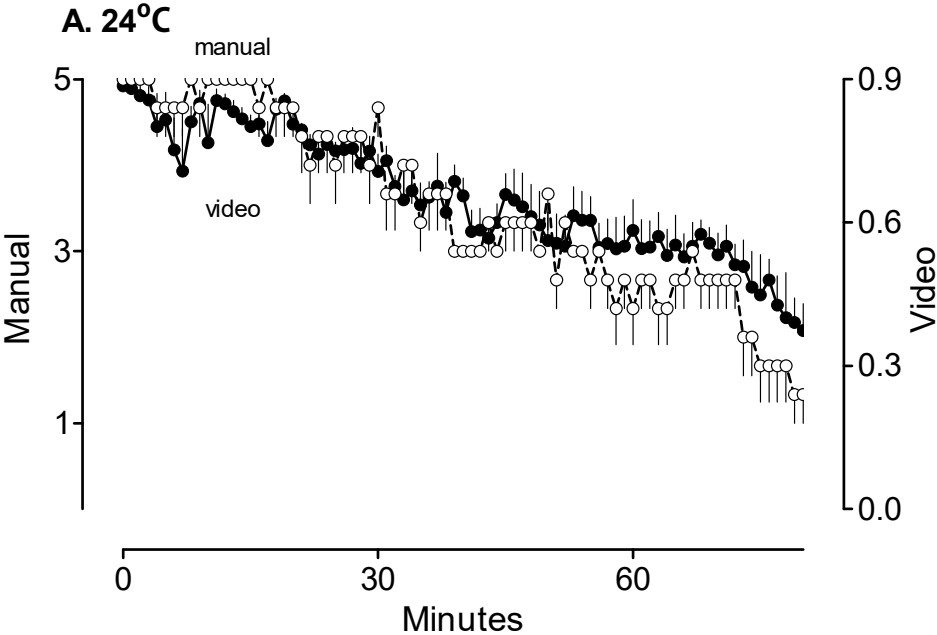

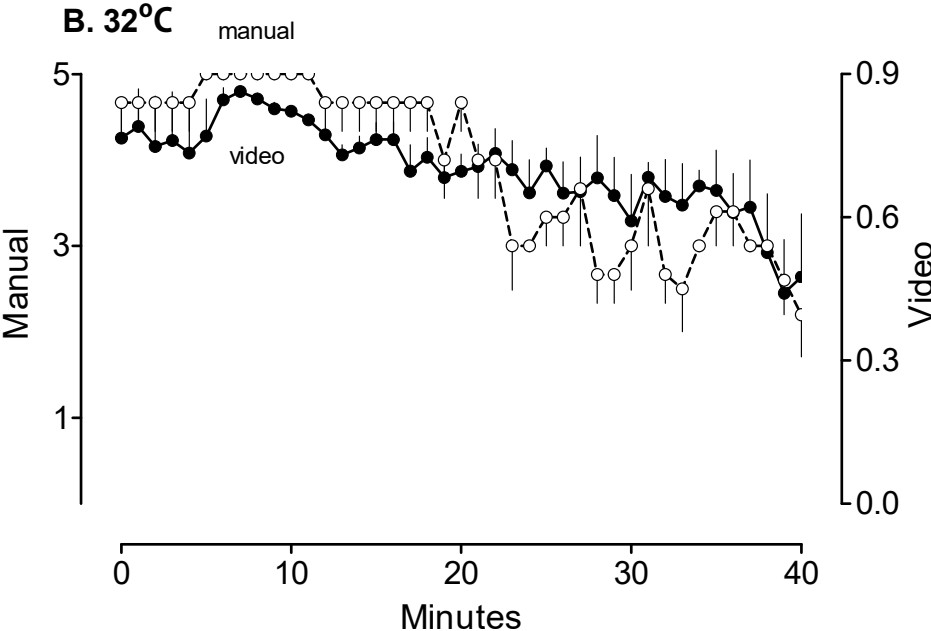

**Figure 3** **The comparison of the results of the manual scoring and automatic analysis of the position on the treadmill.** Rats ran at room temperature (A, 24 °C, N = 6) or in a hot environment (B, 32 °C, N = 6). Averages were calculated as long as at least three rats were still running. Open circles—the results of the observer (manual); closed circles—the results of the video tracking (video).

time to transition from score 3 to score 1 determined by the video tracking and the time to exhaustion was 0.99.

## DISCUSSION

This study outlines a system to objectively measure fatigue and exhaustion in animals running on a treadmill. In animal studies, the major feature of exhaustion is an inability of an animal to keep pace with the treadmill belt despite prodding, by either physical means or an electric shock. There is, however, variability in how researchers define this. For example, some researchers add the requirement that animals must be unable to move for period of time after exhaustion occurs (*Hasegawa et al., 2008*). Likewise, fatigue has been defined as the inability of an animal to keep pace with the treadmill (*Rodrigues et al., 2009*; *Soares et al., 2004*). But how does one objectively measure this?

Our study showed that the subjective grading of running by an observer correlates with the position of an animal on a treadmill as measured by the video tracking software. Therefore, the averaged position of the animal on the treadmill, measured by video tracking software, can provide an objective estimate of the quality of running. Using the same measure, the time at which fatigue and the exhaustion occurs can be estimated.

We tested video analysis technique using rats running on the treadmill in two environmental conditions: at room temperature and in hot environment. Significant differences in times between two environments, the phenomenon which is well-known from studies (*Drummond et al., 2016*) and from personal experience of athletes (*Molkov & Zaretsky, 2016*), allowed for the visualization of descriptive power of automatic analysis.

The data recorded by the human observer clearly correlated with the results of automatic processing, as shown in Fig. 3, in both of these environments. Considering that the parameters of analysis were tuned so that results of the video tracking resemble the estimations of the observer, it is not surprising that the time to fatigue and the time to exhaustion determined by both the observer and automatic analysis were not different.

Both approaches revealed that the hot environment statistically significantly decreases the times to both fatigue and exhaustion when compared with the same times at room temperature. The difference between two environments can be interpreted using known core body temperature dynamics. Hyperthermia suppresses the ability to perform physical work. Exercise-related heat production, which drives an increase of body temperature, is not dependent on ambient temperature (*Yoo et al., 2015*). However, in the hot environment the dissipation of heat is less effective, and thermoregulatory responses to exercise are not effective (*Yoo et al., 2015*). Therefore, it takes less time for body temperature to reach thresholds affecting the ability to run. Both the human observer and the video tracking system identified the occurrence of fatigue at earlier times in the hot environment. Importantly, the automatic measurements of the time to fatigue were not statistically different from human estimates in both environments.

In the hot environment, rats became unable to run shortly after showing that they are fatigued. At fatigue, the body temperature of rats is in the range of 39−40 °C. It leaves very little reserve for further increase, while body temperature is raising at 0.1–0.2 °C/min

(*Zaretsky et al., 2015*). In the absence of effective heat dissipation, animals quickly reach the thermal threshold for the exhaustion, which was estimated at 42.2–42.5 °C (*Walters et al., 2000*). A fast rise of body temperature is reflected in a progressive loss of performance with the corresponding average sliding to the back of the treadmill.

In contrast, at room temperature, after rats became fatigued, they still were able to run for a long time. At room temperature, heat loss mechanisms have sufficient power to dissipate the heat generated by exercise at relatively low power which was used in this study (*Tanaka, Yanase & Nakayama, 1988*). After hyperthermia induces cutaneous vasodilation, body temperature stabilizes at elevated levels. As a result, the average position on the treadmill drifts to the end of the treadmill slowly.

As one of the potential advantages of having the record of running quality over time, we found that the time to exhaustion can be predicted before the exhaustion itself occurs. Most researchers consider exhaustion when the animal is not able to move; a subjective score of 0 in our manual grading system. Unfortunately, reaching such a score results in the animal receiving multiple shocks.

We found that the time of transition from running with score 3 to running with score 1 correlates with the time to exhaustion with a correlation coefficient of 0.99. However, the transition occurs a few minutes before actual exhaustion, which allows to reliably and objectively estimate the time to exhaustion, but to spare the animal of the most stressful experience during the test. Importantly, the difference corresponds to the similar percentage of the total time in both environments, which explains such a high correlation.

## CONCLUSION

An objective nature of estimation of times to fatigue and exhaustion has the advantage of eliminating potential observer bias. Importantly, this technique can also dramatically increase the throughput of experiments as multiple animals can be run and recorded at the same time. In addition, universal scoring, performed via automatic means, can standardize the definitions of fatigue and make the studies done by independent groups more comparable. Lastly, this technique may open a way for a more humane way of studying exhaustion. The time to the decrease in the average running position below 0.35 provides a reliable estimate of time to exhaustion, while eliminating the stressful component of repeated shocks. We consider that the use of objective measures of running quality may help to improve animal care in experiments focusing on studying exercise, endurance and exhaustion.

## ACKNOWLEDGEMENTS

The authors thank Dr. William Truitt for generously providing a computer with licensed Anymaze software which was used for video tracking in this study.

### Funding

Research reported in this publication was supported by the National Institute on Drug Abuse of the NIH under award number R01DA026867. Furthermore, this work was conducted in a facility constructed under support from the National Center for Research Resources, of the NIH under award number C06 RR015481-010. The funders had no role in study design, data collection and analysis, decision to publish, or preparation of the manuscript.

### Grant Disclosures

The following grant information was disclosed by the authors:
National Institute on Drug Abuse of the NIH: R01DA026867.
National Center for Research Resources, of the NIH: C06 RR015481-010.

### Competing Interests

The authors declare there are no competing interests.

### Author Contributions

- Dmitry V. Zaretsky, Hannah Kline, Maria V. Zaretskaia and Daniel E. Rusyniak conceived and designed the experiments, performed the experiments, analyzed the data, contributed reagents/materials/analysis tools, prepared figures and/or tables, authored or reviewed drafts of the paper, approved the final draft.

### Animal Ethics

The following information was supplied relating to ethical approvals (i.e., approving body and any reference numbers):

All procedures were approved by the Indiana University Animal Care and Use Committee (#10093).

### Data Availability

The raw data and code are provided in the Supplemental Files.

### Supplemental Information

Supplemental information for this article can be found online at http://dx.doi.org/10.7717/peerj.5017#supplemental-information.

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
