# Peer review of "Automatic analysis of treadmill running to estimate times to fatigue and exhaustion in rodents"

_PeerJ, doi:10.7717/peerj.5017_

## Round 0.1 · original submission · Major Revisions

As recommended by the reviewers, please strengthen the introduction and the discussion in order to position the study in context of the previous work in the field.

·

Basic reporting

1) The manuscript is written in a clear, unambiguous and technically correct English. The manuscript also conforms to professional standards of courtesy and expression. I have a minor comment regarding language: the authors wrote “time to fatigue and exhaustion” in some sentences, but “times to fatigue and exhaustion” in other sentences. I suggest a standardized use of this phrase throughout the manuscript. In addition, this phrase was repeated 9 times in the abstract; the readability of this section would improve if authors avoid unnecessary repetition.

2) Relevant prior literature was appropriately referenced. However, the manuscript does not include sufficient introduction and background to demonstrate how the work fits into the broader field of knowledge. The manuscript lacks a clear theoretical dissociation between fatigue and exhaustion that could be presented already in the introduction.

3) The structure of the manuscript conforms to an acceptable format of standard sections. Figures are relevant to the content of the article, of sufficient resolution, and appropriately described and labeled. All appropriate raw data were made available in accordance with PeerJ Data Sharing policy.

4) The submission represents an appropriate unit of publication and includes all results relevant to the hypothesis. It seems to me that coherent bodies of work were not inappropriately subdivided merely to increase publication count.

Experimental design

1) The manuscript consists of original primary research within Aims and Scope of the journal. The topic addressed in the present study can classified into the following categories: Biological Sciences and Environmental Sciences.

2) The submission clearly defines the research question, which is relevant and meaningful. The knowledge gap being investigated was identified, and statements were made as to how the study contributes to filling that gap.
3) The investigation was conducted rigorously and to a high technical standard. Moreover, the research was conducted in conformity with the prevailing ethical standards in the field.

4) Methods are described with sufficient information to be reproducible by another investigator.

Validity of the findings

1) The manuscript does not present pointless repetition of well known, widely accepted results.

2) The data are robust, statistically sound, and controlled. I have a minor question about statistics that is written in the “General comments for the author”.

3) The conclusions are appropriately stated, connected to the original question investigated, and limited to those supported by the results.

4) The authors did not discuss their findings in a speculative way.

Additional comments

Zaretsky and colleagues developed an objective method to estimate times to fatigue and exhaustion in rats subjected to treadmill running. The authors hypothesized that the change in the time-averaged position of a rat on a treadmill could be used to correctly estimate times to fatigue and exhaustion. To test this hypothesis, the authors compared the times to fatigue and exhaustion measured by a human observer to times measured by an automated video tracking system. Rats were subjected to treadmill running at either room temperature (24oC) or in hot environment (32oC). The estimates of times made by the observer and the automated video tracking were not statistically different from each other. In addition, both the observer and the automated video tracking determined that exercise in a hot environment, compared with the exercise at room temperature, resulted in shorter times to exhaustion and fatigue, as expected. I have the impression that Zaretsky and colleagues have succeeded in their attempt to develop an objective method that estimates times to fatigue and exhaustion in running rats.

The topic addressed by the present study is relevant, and the findings are very concise and clear. Of note, the development of an objective method to measure times to fatigue and exhaustion is a substantial achievement by the present study, which may greatly influence future studies on this topic. Despite the aforementioned strengths, the authors should improve their paper before publication; particularly, a clear theoretical dissociation between fatigue and exhaustion must be presented already in the introduction. I am also presenting some additional comments aimed at improving the present paper as follows.

Major points:
1. The manuscript does not define the terms fatigue and exhaustion. Although this is a major limitation in the exercise physiology literature, a clear theoretical model explaining the differences between the two terms is important to support the different objective criteria used to determine fatigue and exhaustion.

1.1. These definitions can be very simple, but they will improve a lot the clarity of the manuscript. For example, is exhaustion a condition of extreme fatigue?

1.2. When should a researcher measure time to fatigue instead of time to exhaustion? Do the measures of times to fatigue and exhaustion provide different information? More specifically, when is it relevant to measure time to fatigue? Are there different physiological mechanisms underlying fatigue and exhaustion?

Minor points:

1. Abstract, methods: what was the criterion used to determine that the observer was trained? For how many years has this trained observer been performing experiments with rats subject to treadmill running?

2. Methods: when developing a new method, the authors should quantify the test-retest reliability of their results. In this context, it would be very important if authors calculate the reliability and the intraclass correlation coefficient of the time to fatigue or time to exhaustion when using their recently developed method. For this purpose, it is useful to determine fatigue and exhaustion in a same rat at a same environment during two different exercise sessions on separate days. These analyses are important to determine the minimal difference needed to be exhibited for one to be confident that a true change in performance of an animal has occurred [Weir. J Strength Cond Res, v.19 (n.1): p.231–240, 2005].

3. Methods, lines 103 – 122: was the 5-day familiarization protocol sufficiently long to ensure that rats would run steadily on a treadmill? Could the rats present a more steady performance if they were subjected to additional familiarization sessions?

4. Methods, lines 111 – 112: the authors should present the intensity in amperes (not only the frequency) of the mild electric stimuli used during the familiarization sessions and experimental trials.

5. Methods: the information about relative humidity and ambient temperature was written in two different sections of the manuscript (lines 90 – 91 and lines 117 – 118). The authors should avoid unnecessary repetition of information.

6. Methods, lines 121 – 123: the authors wrote that “When a rat was consistently unable to maintain the pace of the treadmill and received three continuous electric shocks on the grid at the back of the treadmill, the belt was stopped”. Is this information correct? Did the trained observer use this criterion to determine exhaustion?

7. Methods, lines 141 – 153: the authors did not describe the meaning / characteristics of treadmill running associated with scores 2 and 4? Why didn’t the authors use these two scores?

8. Methods, lines 249 – 250: the authors wrote that: “In the first 15 min of recording, the lowest points on the curve do not fall below 0.15 m”. Is the 0.15 value a normalized position on the treadmill?

9. Methods, lines 266 – 269: I felt that the information in the following two sentences is not coherent (average coefficient of 0.75 vs. was exactly the same). “Manual scores and average positions correlated significantly with an average coefficient of 0.75±0.03. The time to reach averaged position close to zero was exactly the same as the time of exhaustion determined by the observer in the case of each rat.” Please amend these sentences to make them more coherent.

10. Results, lines 295 – 297: the authors wrote that “At room temperature, exhaustion developed 58±5 min after fatigue if determined by the observer or 52±2 min if determined by the video tracking (p>0.05)”. Why did the authors determine time to exhaustion after fatigue? These results would be much clearer if times to exhaustion had been determined from exercise initiation.

11. Discussion, lines 320 – 321: the authors wrote that “We tested video analysis technique using rats running on the treadmill in two environmental conditions: at room temperature and in hot environment”. The fact that aerobic performance was degraded in the heat is not a novel finding provided by the present study. The authors should state in the methods or discussion that performance was compared between the two environments to test the sensitivity of their new method. Previous investigations have indicated that aerobic performance is lower in the heat than at room temperature, including studies using very similar ambient temperatures to those used in the present study [Drummond et al. PLOS One v.11 (n.5): e0155919, 2016].

12. Discussion, lines 350 – 352: the authors wrote that “At room temperature, exhaustion is dependent on multiple inputs, e.g. dehydration, interacting with thermoregulation”. Dehydration is not an expected outcome in exercising rats, as they do not sweat to regulate their temperature. The authors may want to suggest the involvement of other physiological responses that could underlie the development of exhaustion.

·

Basic reporting

This is an interest paper about the objective determination of fatigue in rodents. The authors have done a nice job describing the methods to perform an automatic analysis of time to fatigue in running rats. This study is an important contribution to experiments using animals in exercise physiology science. I have included some specific comments that the authors should consider.
Introduction
The authors can improve the introduction of the paper by describing previous studies that measured the total exercise time in rodents. It is important for the reader to understand how this measure has been made in order to understand the novelty of the proposed method.
The authors should also mention in the introduction the theoretical basis of the effects of heat on fatigue.
It is important to explain that the rats run through an electrical stimulus and the magnitude of this can drastically influence the total exercise time.
Line 72: please include in rodents … and measure time to fatigue and exhaustion in rodents.

Experimental design

Methods
Is the sample size adequate? The total exercise time in running rats is a variable that presents a high coefficient of variation. Generally, a sample size greater than 6 is required.
Experimental protocol
Line 112. How many milliamps and how many millivolts were applied to the electric stimulus?
Line 114. Please change “treadmill running” to “constant-speed exercise”
Authors should provide further details of manual determination of total exercise time. For example, some authors establish as a criterion for the interruption of effort when the rat remains in the electric stimulus grid for more than 10 seconds.

Results
It is important for the authors to describe in graphs or tables the correlation coefficient between the time to fatigue values reached by the manual method and by the automatic method.
Is the automatic analysis specific for the Columbus treadmill? On the other hand, the Excel spreadsheet can be modified for use with other apparatus. There is variation in the size of the belt between different treadmills.
The authors should present a graph with the internal temperature values of the rats during running at 24°C and 32°C. This will contribute to the discussion of the effects of heat on fatigue.
Line 286. Delete “…results in faster fatigue and exhaustion” and maintain Exercise in a hot environment

Paragraph 1 – If the automatic analysis does not differ from the manual, how important is the use of automatic analysis? The authors should explain this in the first paragraph of discussion.

Validity of the findings

The results presented support the conclusion of the authors. Statistical analysis was well done. The papere will gain in quality with the presentation of correlations and internal temperature values of rats.

Additional comments

Many previous studies have evaluated exercise time to fatigue in rats. Perhaps the present paper will be better if the authors compare the data obtained with these previous studies in the discussion session.

---

## Round 0.2 · accepted · Accept

Thank you for carefully addressing the issues brought forward by the reviewers.

# ·

Basic reporting

No comment.

Experimental design

No comment.

Validity of the findings

No comment.

Additional comments

Zaretsky and colleagues have answered my comments in a satisfactory manner. Indeed, they have improved the introduction section, which now provides a strong scientific rationale that conducts the readers to the main problem addressed by the study. The definitions of the terms “fatigue” and “exhaustion” were presented in the revised introduction, and the authors provided clear examples that support these definitions. The revised manuscript is much better than the original one. Congratulations!

·

Basic reporting

The authors accepted the suggestions of the reviewers about english, article structure and background.

Experimental design

The first version of the manuscript already presented adequate methodological description and the authors accepted the main suggestions of the reviewers about the details of the experimental design.

Validity of the findings

The methodological advances reached in the present study will contribute to the study of fatigue mechanisms in experiments involving animals.

Additional comments

My main suggestions have been accepted by the authors, so I agree with the publication of the study.